# The Role of PPARβ/δ-Related Lipid Metabolism in High-Altitude Adaptation of Yak Coat Based on Proteomics and Metabolomics

**DOI:** 10.3390/cells14231843

**Published:** 2025-11-22

**Authors:** Shijie Li, Yan Cui, Xinrui Li, Xuefeng Bai, Denghui Wang, Pengfei Zhao, Pengqiang Wei, Sijiu Yu, Junfeng He

**Affiliations:** 1Laboratory of Animal Anatomy & Tissue Embryology, Department of Basic Veterinary Medicine, Faculty of Veterinary Medicine, Gansu Agricultural University, Lanzhou 730070, China; shijieli126@126.com (S.L.); 107331902083@st.gsau.edu.cn (X.L.); 107332001029@st.gsau.edu.cn (X.B.); 1073323020121@st.gsau.edu.cn (D.W.); 107332201037@st.gsau.edu.cn (P.Z.); weipengqiang@gsau.edu.cn (P.W.); hejf@gsau.edu.cn (J.H.); 2Gansu Province Livestock Embryo Engineering Research Center, Department of Clinical Veterinary Medicine, Faculty of Veterinary Medicine, Gansu Agricultural University, Lanzhou 730070, China; sijiuy@126.com

**Keywords:** PPARβ/δ, lipid, hair follicle, cycle

## Abstract

**Background:** In the cold plateau environment, the yak’s coat exhibits significant adaptive regulation to cope with adverse conditions. This adaptation is fundamentally governed by the cycle of hair follicles (HFs), a complex process involving numerous molecular signals. However, the key regulators and underlying pathways remain poorly understood. **Methods:** Proteomic and non-targeted metabolomic analyses were employed to systematically investigate changes in proteins and metabolites during the cycle of yak hair follicles. We further validated the expression dynamics of PPARβ/δ and its related molecules, as well as the specific biological role of PPARβ/δ in regulating lipid metabolism and influencing the proliferation and apoptosis of yak dermal papilla cells (DPCs). **Results:** Proteomic results indicated that lipid-related proteins were among the most significantly altered, second only to hair structural proteins. The PPAR signaling pathway, which regulates lipid metabolism, may also play an important role in the cycle of yak HF. Non-targeted metabolomics revealed that Fatty Acyls were the most significantly altered metabolites during the transitions into anagen and catagen. Notably, unsaturated long-chain fatty acids (PPARβ/δ agonists) were consistently up-regulated in anagen and down-regulated in catagen, whereas saturated long-chain fatty acids (lacking PPARβ/δ agonist activity) did not exhibit a similar trend. PPARβ/δ shows significant expression changes in the dermal papilla (DP) and hair matrix (HM) during the cycle of yak HFs. Specifically, PPARβ/δ expression in the DP underwent progressive downregulation during the transition from anagen to catagen and subsequently to telogen, becoming nearly undetectable in the telogen DP. Cellular experiments confirmed that PPARβ/δ activation significantly reduced intracellular lipid content in yak DPCs and was accompanied by increased proliferation. Conversely, PPARβ/δ inhibition led to intracellular lipid accumulation and decreased proliferation. **Conclusions:** These findings suggest that PPARβ/δ may regulate the yak HF cycle by modulating lipid metabolism in DP. The level of intrinsic lipid metabolism within HFs may be a key factor influencing yak HF growth.

## 1. Introduction

The plateau represents a unique climatic environment for animal survival, characterized by multiple extreme environmental factors that pose significant challenges to its inhabitants [1]. Adaptation to such harsh conditions often requires specialized physiological mechanisms [2]. As a typical plateau species, the yak inhabits regions at elevations above 2500 m and exhibits a distinct periodic growth pattern of its coat, which plays a crucial role in its adaptation to the cold, intense ultraviolet radiation, and large seasonal temperature fluctuations typical of high-altitude environments [3,4,5]. Changes in coat thickness are driven by the synchronized cyclic growth of a large number of hair follicles (HFs), which undergo clearly defined phases of anagen, catagen, and telogen. This well-characterized follicular cycling makes the yak an ideal animal model for studying periodic HF growth [2,4].

In proteomic studies on the cycle of yak HF, the PPAR signaling pathway was found to be significantly enriched during the transition of HFs into anagen and catagen, with marked differences in the expression of PPARβ/δ and numerous lipid-related proteins. Peroxisome proliferator-activated receptors (PPARs) are transcription factors belonging to the nuclear hormone receptor superfamily, including PPARα, PPARγ, and PPARβ/δ [6]. They primarily regulate lipid-related metabolism in both adipose and non-adipose tissues and function as ligand-activated transcription factors [7]. PPARβ/δ is widely distributed across various tissues and organs, with high expression in the colon, small intestine, liver, and skin [8]. Notably, its expression in the skin is significantly higher than that of the other two subtypes [9,10]. Studies have shown that unsaturated fatty acids and retinoic acid are important ligand activators of PPARβ/δ [11,12]. Research on PPARβ/δ in the skin has primarily focused on the epidermis and its mesenchyme, involving pathological models such as psoriasis, wound healing, and impaired atopic dermatitis [8,13]. PPARβ/δ has been demonstrated to promote the proliferation of epidermal keratinocytes and play a positive role in wound healing [10,14]. Additionally, upregulated PPARβ/δ expression has been observed in the skin lesions of psoriasis and impaired atopic dermatitis, and inhibiting PPARβ/δ may offer therapeutic potential for diseases associated with hyperproliferation of epidermal keratinocytes [15,16,17].

The hair follicle, a skin appendage, is composed of an epithelial component (hair matrix HM) and a mesenchymal component (dermal papilla DP), analogous to the epidermis and dermis, respectively. The growth of hair follicles depends on the communication between the DP and HM, as well as the dynamic balance of their respective proliferation and apoptosis [18,19]. Research on PPARβ/δ in hair follicles is actually not extensive and mainly focuses on the developmental stage of HFs. Studies have shown that PPARβ/δ plays an important role in hair growth, as this receptor is highly expressed in follicular keratinocytes throughout the process of HF morphogenesis [20]. PPARβ/δ is particularly important during the developmental stage of hair peg elongation and protects elongating HFs from apoptosis via activation of the Akt1 signaling pathway [20]. In addition, epithelium–mesenchyme interactions in the skin regulate the activity of PPARβ/δ by controlling the production of its endogenous ligands during HF development, which in turn regulates cell proliferation and differentiation, metabolism, vascular growth, and inflammation [21]. However, the specific roles of PPARβ/δ in the epithelial and dermal components of HFs have not yet been reported. To investigate whether yaks in the high-altitude cold environment influence lipid metabolism through PPARβ/δ and thereby regulate the cycle of HF, we explored its role in the periodic growth of yak HF, aiming to provide new insights for HF medicine and research on the adaptation of plateau animals.

Schematic figure, Figure 1:

## 2. Materials and Methods

### 2.1. Animals

Back skin samples (1 cm × 2 cm) were longitudinally collected from five healthy, two-year-old male yaks in January, June, and November of the same year from the pastoral area of Tianzhu County (altitude: 3000 m). To control for inter-individual variation, all samples from different time points were obtained from the same individuals. Upon collection, the tissues were rinsed with normal saline, blotted dry with absorbent paper, and subsequently partitioned for various analyses, including histology, primary cell culture, metabolomics, and proteomics. All animal procedures were approved by the Animal Care and Use Committee of Gansu Agriculture University (Ethics Approval No. GSAU-Eth-VMC-2023-004).

### 2.2. HE Staining of Yak HF in Different Growth Stages

Yak skin tissues pre-fixed in 4% paraformaldehyde (PFA) were embedded in paraffin and sectioned into 5 μm-thick slices. The sections were routinely dewaxed and stained with hematoxylin and eosin ( Solarbio, Beijing, China). Ten intact follicular units at different HF growth stages were randomly selected. With the primary HF as the central structure, the number of surrounding secondary HF and the diameters of the follicles were quantified. All measurements and quantitative analyses were performed using ImageJ software (version 1.54).

### 2.3. Primary Cell Culture of Yak DPCs

Yak skin tissue was cut into tissue slices with a size of 1 cm × 1 cm and a thickness of less than 1 mm along the hair shaft direction. Intact HF were bluntly separated under a stereomicroscope and incubated in 0.1% collagenase II at 37 °C for 2 h. DP was finely separated and transferred into 10% FBS complete culture medium (DMEM added with 1% penicillin and streptomycin). DP was fixed at the bottom of the dish by a sterile needle and cultured in a 37 °C and 5% CO_2_ atmosphere to obtain yak primary DPCs.

### 2.4. Proteomics and DEPs Analysis of Yak Hair Cycling

Proteomic analysis was performed on yak skin tissue samples from three different HF growth periods. The steps of the protein extraction, SDS-PAGE, protein digestion and TMT labeling have been described in detail in previous studies [22]. The resulting LC-MS/MS data were processed using the Thermo Proteome Discoverer (v.2.4). Tandem mass spectra (Rigol L-3000, Suzhou, China) were searched against Bos taurus (37,513 sequences) from the UniProt database (http://beta.uniprot.org/, accessed on 25 May 2021). Differential expression analysis was executed under stringent criteria: adjusted *p*-value (FDR) ≤ 0.05 and absolute fold change ≥ 1.2. The Kyoto Encyclopedia of Genes and Genomes (KEGG) database was used to annotate metabolic pathways. Interactions between proteins were evaluated with the STRING database (http://string-db.org/, accessed on 25 May 2021) based on a minimum interaction score cut-off of 0.400. The software used for figures is Graphpad prism (10.1.2) and Cytoscape (3.9.1).

### 2.5. Metabolomics and Differential Metabolites (DMs) Analysis of Yak Hair Cycling

Yak skin samples, corresponding to those used in proteomic analysis, were subjected to untargeted metabolomic profiling. For each sample, 50 mg of tissue was accurately weighed and transferred to a 2 mL centrifuge tube. Then, 1000 µL of extraction solvent (consisting of 75% methanol: chloroform mixture [9:1, *v*/*v*; methanol from Thermo Fisher Scientific, Waltham, MA, USA; chloroform from Sinopharm, Beijing, China] and 25% H_2_O) along with three steel beads were added. The samples were homogenized using a tissue grinder at 50 Hz for 60 s; this grinding procedure was repeated twice. Subsequently, the samples were ultrasonicated at room temperature for 30 min, followed by incubation in an ice bath for another 30 min. After extraction, the mixtures were centrifuged at 12,000 rpm and 4 °C for 10 min. The entire supernatant was collected, transferred to a new 2 mL centrifuge tube, and then concentrated to complete dryness. The dried metabolites were reconstituted in 200 µL of 50% acetonitrile (Thermo Fisher Scientific, Waltham, MA, USA) containing 4 ppm 2-Amino-3-(2-chlorophenyl)-propionic acid (Aladdin, Shanghai, China) as an internal standard (stored at 4 °C). Finally, the supernatant was filtered through a 0.22 μm membrane and transferred to LC-MS vials for liquid chromatography–mass spectrometry analysis [23]. The LC analysis was performed on a Vanquish UHPLC System (Thermo Fisher Scientific, Waltham, MA, USA). Chromatography was carried out with an ACQUITY UPLC^®^ HSS T3 (150 × 2.1 mm, 1.8 µm) (Waters, Milford, MA, USA). Mass spectrometric detection of metabolites was performed on Orbitrap Exploris 120 (Thermo Fisher Scientific, Waltham, MA, USA) with an ESI ion source. Simultaneous MS1 and MS/MS (Full MS-ddMS2 mode, data-dependent MS/MS) acquisition was used [24]. The metabolites were identified by accurate mass (<30 ppm) and MS/MS data, which were matched with HMDB (http://www.hmdb.ca, accessed on 20 March 2025), massbank (http://www.massbank.jp/, accessed on 20 March 2025), LipidMaps (http://www.lipidmaps.org, accessed on 20 March 2025), Mzclound (https://www.mzcloud.org, accessed on 20 March 2025) and KEGG (http://www.genome.jp/kegg/, accessed on 20 March 2025). Finally, *p* value < 0.05 and VIP value > 1 were considered to be statistically significant metabolites.

### 2.6. Immunofluorescence

Paraffin-embedded yak skin tissues were sectioned at a thickness of 5 μm. Following deparaffinization, the sections were permeabilized with 0.1% Triton X-100 (Beyotime, Shanghai, China) for 10 min and then blocked with 5% BSA for 1 h at room temperature. The sections were incubated overnight at 4 °C with primary antibodies (Appendix A) diluted in antibody diluent. After thorough washing with PBST (PBS containing 0.1% Tween-20), the samples were treated with Alexa Fluor 488-conjugated anti-rabbit secondary antibody (1:1000; Cell Signaling Technology, Danvers, MA, USA) for 1 h in the dark. Nuclei were counterstained with DAPI (10 μg/mL) for 5 min, followed by additional PBS washes. For cellular immunofluorescence, DPCs grown on sterile glass coverslips were fixed with 4% paraformaldehyde for 15 min at room temperature, after which the aforementioned staining protocol was followed. All images were captured using an Olympus-DP71 fluorescence microscope (Olympus, Tokyo, Japan). Quantitative analysis was performed on six randomly selected fields per slide using ImageJ 1.54 software (NIH, Bethesda, MD, USA), applying background subtraction and thresholding.

### 2.7. Western Blotting

Total protein extraction from skin tissues and DPCs was performed using RIPA lysis buffer (Solarbio, Beijing, China), supplemented with 1% protease inhibitor cocktail and 1 mM PMSF. The protein extracts were then resolved on 10% polyacrylamide gels via SDS-PAGE, employing a Bio-Rad Mini-Protean Tetra system at 220 V for 40 min. Following separation, the proteins were electrophoretically transferred to polyvinylidene difluoride (PVDF) membranes (Millipore, Darmstadt, Germany). To block non-specific binding sites, the membranes were incubated with 5% nonfat milk in PBS for 2 h at room temperature. Subsequently, the membranes were probed with specific primary antibodies (Appendix A) in an overnight incubation at 4 °C, followed by a 50 min incubation with appropriate secondary antibodies at room temperature. Protein bands were visualized with an ECL reagent (Beyotime, Shanghai, China) and imaged using a Tanon 5200 Chemiluminescent Imaging System (Shanghai, China). Finally, band intensities were quantified with ImageJ 1.54 software (NIH, Bethesda, MD, USA) after background subtraction, normalizing to GAPDH as a loading control.

### 2.8. Cell Proliferation EDU Assay

DPCs were cultured in 12-well plates for 24 h. Cell proliferation was assessed using an EdU Cell Proliferation Kit (Alexa Fluor 488, Beyotime, Shanghai, China) according to the manufacturer’s instructions. Briefly, after a 2 h pulse with 50 μM EdU at 37 °C, the cells were fixed with 4% paraformaldehyde and permeabilized with 0.1% Triton X-100. The incorporated EdU was then labeled via a 30 min click reaction catalyzed by CuSO_4_ (10 μM) in the presence of APC azide (2 μM) and ascorbic acid (1 mM). Cell nuclei were counterstained with Hoechst 33,342 (5 μg/mL) for 10 min. Fluorescence imaging was performed using an DP71 microscope (Olympus, Tokyo, Japan), with EdU and Hoechst signals detected under 488 nm and 405 nm lasers, respectively. Image analysis, including automated thresholding and intensity quantification, was conducted using ImageJ software (version 1.54).

### 2.9. Cell Scratch Assay

DPCs were cultured in 6-well plates until the cells reached 80–90% confluency. The cells were then serum-starved for 8 h. A uniform wound was created in each well using a sterile cell scraper and a ruler as a guide. The dislodged cells were removed by washing with PBS. The plates were subsequently incubated at 37 °C under 5% CO_2_ for 48 h. Images of the wound area were captured at the same location at 0, 12 and 24 h post-scratching. Cell images were acquired using an DP71 fluorescence microscope (Olympus, Tokyo, Japan). ImageJ software (version 1.54) was employed to analyze the scratch area images.

### 2.10. Drug Treatment

DPCs were cultured in six-well plates and treated with the PPARβ/δ agonist GW0742 (MCE, Shanghai, China) or antagonist DG172 dihydrochloride (MCE, Shanghai, China) for 24 h. Based on the drug manufacturer’s instructions and CCK-8 assay results, which indicated no significant cytotoxicity within the tested concentration range, the optimal treatment concentrations were determined as follows: 10 μM GW0742, 10 μM DG172 dihydrochloride for DPCs. The experimental groups were designed as follows: (1) control group: cells treated with PBS instead of the drug; (2) experimental groups: cells treated with different drug concentrations, yielding a total of four experimental conditions.

### 2.11. CCK-8 Cytotoxicity Assay

DPCs were seeded in 96-well plates and, after adhesion, were subjected to serum starvation for 8 h. Based on previous literature [25], the cells were treated with the PPARβ/δ agonist GW0742 or antagonist DG172 dihydrochloride across a concentration range (0.01, 0.1, 1, 5, 10, 50, and 100 μM). The experimental setup included two control groups: a blank control (CCK-8 reagent with culture medium) and a zero-drug control (cells with culture medium and CCK-8 reagent). Following the manufacturer’s instructions, the CCK-8 assay was conducted to determine cell viability, which was calculated as: Cell viability (%) = [A(drug-treated) − A(blank)]/[A(zero-drug) − A(blank)] × 100%. Subsequent statistical analysis and data visualization were performed using GraphPad Prism (version 10.1.2).

### 2.12. Statistics Analyses

Statistical analyses were performed using GraphPad Prism (version 10.1.2). All experiments were conducted with at least three independent replicates. Data are presented as mean ± standard error of the mean (SEM). Comparisons between two groups were analyzed using Student’s *t*-test, while one-way analysis of variance (ANOVA) followed by an appropriate post hoc test was applied for comparisons among multiple groups. A *p*-value of less than 0.05 was considered statistically significant.

## 3. Results

### 3.1. Characteristics of Yak HF Cycling

To elucidate the pattern of cyclical HF growth in yaks, we compared the distribution and histological morphology of HF in anagen, catagen and telogen. HE staining revealed that secondary hair follicles (SFs) within the anagen HF clusters were markedly developed. The number of SFs surrounding the primary hair follicles (PFs) was significantly higher in anagen than in catagen and telogen, while the difference between the catagen and telogen was minimal, indicating a progressive reduction in follicle number during the transition from anagen through catagen to telogen (Figure 1A,B). Significant differences in HF size were observed across the various growth phases. The average diameter of PFs was largest in telogen, followed by catagen, and smallest in anagen. A similar trend was observed for the average diameter of SFs, suggesting a gradual enlargement of follicles as growth transitions from anagen through catagen to telogen (Figure 1A,C). These results demonstrate that the growth of yak HF is a dynamic process, with distinct characteristics defining each phase of the HF cycle.

### 3.2. Proteomic Analysis of Yak HF Cycling

To investigate the key factors regulating yak HF growth, proteomic analysis was performed on skin samples collected during anagen, catagen and telogen. The results revealed significant differences in protein expression among the three stages (Figure 2A). Specifically, 550 DEPs were identified between telogen and anagen, with 230 upregulated and 320 downregulated. Between catagen and anagen, 430 DEPs were detected, comprising 232 upregulated and 198 downregulated. Additionally, 281 DEPs were found between telogen and catagen, with 176 upregulated and 105 downregulated (Figure 2B).

Functional enrichment analysis of these DEPs indicated significant alterations in lipid-related proteins and pathways, particularly during yak HF growth and regression. The Top20 KEGG enrichment results showed that among the upregulated DEPs associated with HF growth, the PPAR signaling pathway was the most significantly enriched. Pathways such as unsaturated fatty acid biosynthesis, fat digestion and absorption and fatty acid elongation were also notably enriched (Figure 2C). Among the downregulated proteins, the fat digestion and absorption signaling pathway was significantly enriched (Figure 2D). For the upregulated DEPs related to HF regression, the adipokine signaling pathway was prominently enriched (Figure 2E). Notably, among the downregulated proteins, the unsaturated fatty acid biosynthesis pathway, which is closely associated with PPAR signaling, was the most significantly enriched. The PPAR signaling pathway, along with fatty acid elongation and fatty acid biosynthesis signaling pathways, was also significantly enriched (Figure 2F). Protein–protein interaction (PPI) network analysis of the top 100 DEPs related to yak HF growth and regression revealed that, in addition to various Krt proteins and cell junction proteins such as DSG1, DSG2, DSP, DSC1, and CLDN1 directly associated with hair structure, several lipid-related proteins including ACSL1, C3, APOH, AUP1, and FASN served as core nodes in the interaction network (Figure 2G,H). These findings suggest that lipid metabolism within the HF or its surrounding microenvironment undergoes significant changes during alterations in yak HF growth states, and the PPAR signaling pathway may play a key role in regulating yak HF growth. DEPs identified in the PPAR signaling pathway include FABP4, FABP5, PPARβ/δ, RXRG, ACAA1, EHHADH, APOA1, ACOX1, HMGCS1, ACSL1, ACSL3, and PLIN2, among which PPARβ/δ is one of the core regulatory factors in this pathway.

### 3.3. Metabolomics Analysis of Yak HF Cycling

The altered expression of a substantial number of lipid-related proteins suggests that lipid metabolism may be a key factor influencing HF growth in yaks. Therefore, we conducted an untargeted metabolomic analysis on yak HF cycling. PLS-DA results under both positive and negative ion modes indicated stable metabolic levels across various growth phases of yak HF (Figure 3A). Cluster analysis revealed distinct metabolic differences among the anagen, catagen and telogen (Figure 3B). Enrichment analysis of the DMs based on chemical structures showed that Fatty Acyls were the most significantly altered metabolic category during the transition to anagen and catagen (Figure 3C,D). In contrast, Carboxylic acids and derivatives became the most prominent category during telogen (Figure 3E). A total of 31 Fatty Acyl metabolites (Appendix A) were identified between telogen and anagen, including 4 saturated long-chain fatty acids and 5 unsaturated long-chain fatty acids (Figure 3C). Between the anagen and catagen, 30 Fatty Acyl metabolites (Appendix A) were identified, also comprising 4 saturated and 5 unsaturated long chain fatty acids (Figure 3D). Interestingly, unsaturated long-chain fatty acids consistently exhibited upregulation during the transition from telogen to anagen and downregulation during the transition to catagen, while saturated long-chain fatty acids did not show similar changes (Figure 3F,G). These results indicate that PPARβ/δ function is likely activated during the anagen, suppressed during the catagen and involved in regulating the cycle of yak HF.

### 3.4. Expression of the DEPs in the PPAR Signaling Pathway in Yak HF Cycling

To elucidate the relationship between the yak HF cycling and lipid metabolism, we compared the expression changes of PPARβ/δ and its upstream and downstream DEPs in the PPAR signaling pathway. It is known that FABPs can bind unsaturated long-chain fatty acids to activate PPARβ/δ, thereby regulating the expression of various downstream lipid-related proteins (Figure 4A). The upstream transporter FABP4 was upregulated in anagen and downregulated in catagen, while FABP5 exhibited the opposite expression pattern (Figure 4B,C). The nuclear receptor proteins PPARβ/δ and RXRG were upregulated in anagen and downregulated in catagen (Figure 4B,D). Downstream target proteins, including PLIN2, ACSL3, ACSL1, ACAA1, and EHHADH, were lowly expressed in anagen and highly expressed in catagen (Figure 4B,E,F). In contrast, APOA1 and HMGCS1 were highly expressed in anagen and lowly expressed in catagen, while ACOX1 showed no significant expression difference between anagen and catagen (Figure 4B,E,F). Notably, the differential expression of the lipid droplet marker protein PLIN2 and the rate-limiting enzymes for lipid synthesis, ACSL1 and ACSL3, suggests that lipid accumulation levels may be low in yak skin or HF in anagen and high in catagen.

The expression patterns of PPARβ/δ and unsaturated long-chain fatty acids in metabolomic analyses indicate that the PPAR signaling pathway may be activated during anagen and suppressed during catagen and telogen. To investigate whether this pathway is associated with the growth and regression of yak HF, we examined the expression of PPARβ/δ using immunofluorescence. The results showed that PPARβ/δ was expressed in the DP, HM and outer root sheath of yak HF, with the most significant expression changes occurring in the DP and HM (Figure 5A–C). PPARβ/δ was strongly expressed in both the DP and HM during anagen (Figure 5A,D). Its expression continuously decreased in the DP during the transition from catagen to telogen (Figure 5B–D), while no significant change was observed in the HM during catagen (Figure 5B,D). However, expression in the HM was significantly downregulated during telogen (Figure 5C,D). These findings suggest that PPARβ/δ may be a key regulator of HF growth in yaks.

### 3.5. Primary Culture and Characterization of Yak DP Cells (DPCs)

To investigate the specific functions of PPARβ/δ in the DP of yak HF, we isolated and cultured yak DPCs in vitro using the primary culture method. Yak DPe were microscopically dissected from HF and anchored in culture dishes to obtain primary yak DPCs (Figure 6A). These primary DPCs exhibited a typical fibroblast-like morphology, appearing either elongated and spindle-shaped or triangular. After three passages, the cells remained in good condition. To closely mimic in vivo conditions, second-passage DPCs were selected for subsequent cell identification and experiments (Figure 6A). To verify the specificity and purity of the yak DPCs, we detected the expression of the DP cell-specific markers SOX2 and Vimentin using immunofluorescence. The results showed that SOX2 was specifically expressed within the DPe of yak HF, with no expression observed in other regions (Figure 6B). Additionally, both SOX2 and Vimentin were positively expressed in the primary cultured yak DPCs (Figure 6B). These findings confirm the successful cultivation of high-purity primary yak DPCs.

### 3.6. Effects of PPARβ/δ on Lipid Metabolism in Yak DPCs

To investigate whether PPARβ/δ influences the cyclic growth of yak HF by modulating intrinsic lipid metabolism, we treated yak DPCs with the PPARβ/δ agonist GW0742 and the antagonist DG172 dihydrochloride to specifically activate or inhibit PPARβ/δ signaling. CCK-8 assays revealed no significant cytotoxicity at concentrations of 0.01 μM, 0.1 μM, 1 μM, 5 μM, or 10 μM for both compounds. However, marked cytotoxicity was observed at 50 μM and 100 Μm (Figure 7A,B). Based on these findings, the highest non-toxic concentration (10 μM) was selected for subsequent experiments. Western blot analysis demonstrated that activation of PPARβ/δ significantly upregulated the expression of PLIN2 and ACSL1, whereas its inhibition led to their significant downregulation (Figure 7C–E). Furthermore, immunofluorescence staining indicated that PPARβ/δ activation reduced PLIN2 expression in DPCs, with a notable decrease in nuclear PLIN2 localization and clearly discernible nuclear outlines in positively stained cells (Figure 7F,G). In contrast, PPARβ/δ inhibition upregulated PLIN2 expression in both the cytoplasm and nucleus, accompanied by the loss of distinct nuclear outlines in positive cells (Figure 7F,G).

### 3.7. Effect of PPARβ/δ on the Proliferation of Yak DPCs

To determine whether PPARβ/δ, in addition to regulating lipid metabolism in yak DPCs, further influences their proliferation, we conducted relevant assessments. EdU staining results revealed a significant increase in fluorescence intensity after 24 h of PPARβ/δ activation, whereas inhibition of PPARβ/δ led to a notable decrease (Figure 8A,B). Cell scratch assays demonstrated that at both 12 h and 24 h post-activation, the cell proliferation coverage was significantly greater than that in the control and inhibitor-treated groups (Figure 8C,D). In contrast, after PPARβ/δ inhibition, the proliferation coverage was significantly lower than that in both the control and agonist-treated groups at the same time points (Figure 8C,D). Furthermore, Western blot analysis showed that activation of PPARβ/δ significantly upregulated the expression of PCNA and BCL-2, while downregulating BAX expression (Figure 8E,F). Conversely, inhibition of PPARβ/δ resulted in a significant decrease in PCNA expression, a mild reduction in BCL-2, and a marked increase in BAX expression (Figure 8E,F). Taken together, these results indicate that PPARβ/δ not only regulates lipid metabolism but also serves as a critical modulator of DPCs proliferation. The differential expression of PPARβ/δ during various stages of the yak HF cycle may modulate lipid metabolism levels within the DP, thereby influencing HF growth.

## 4. Discussion

The cycle of HF is a complex physiological process, yet this continuous cyclic growth is essential for follicular health. The coat of yak exhibits distinct stage-specific growth characteristics due to their living environment, which provides a valuable model for studying HF growth mechanisms and high-altitude adaptation. Proteomic results indicate that, in addition to cell junction proteins and keratins associated with hair structure, lipid-related proteins emerge as the third most significantly enriched category of DEPs during yak HF anagen and catagen. KEGG analysis further reveals that PPAR signaling and multiple fatty acid-related pathways are significantly enriched among the Top 20 entries, suggesting substantial changes in lipid metabolism-related proteins during the yak HF cycling. The PPAR signaling pathway is a central pathway for the regulation of lipid metabolism in both adipose and non-adipose tissues. Specifically, FABP4, regulated by PPARγ signaling, is upregulated in cancer cells, leading to reduced intracellular lipid droplets and suppressed cell proliferation, thereby promoting lipolysis [26]. FABP5 modulates lipid metabolism by facilitating lipid synthesis and redirecting long-chain fatty acids toward complex lipid formation, serving as a key factor in de novo fatty acid synthesis [27]. Although FABP4 and FABP5, as fatty acid chaperone molecules, exhibit opposing expression trends in yak HF cycling, they may act synergistically in skin lipid metabolism. ACSL1 and ACSL3 are key enzymes in de novo synthesis of long-chain long chain fatty acids and have been demonstrated to promote intracellular lipid accumulation [28,29,30]. PPARβ/δ is the only subtype showing significant differential expression in yak HF cycling, and PPARs are critical regulators in the PPAR signaling pathway of lipid metabolism [7]. The expression of PLIN2, a marker protein for intracellular lipid droplets, which are key organelles for cellular lipid and energy homeostasis, changes during the cycle of yak HF. This alteration provides direct evidence for the extent of lipid accumulation across different growth phases [31]. Based on these findings, it can be inferred that the skin lipid content changes markedly in yak HF cycling, showing peak accumulation in catagen, moderate levels in telogen, and minimal levels in anagen. The PPAR signaling pathway is likely to be a key regulator underlying these dynamic changes.

PPARβ/δ is activated by endogenous ligands including fatty acids and their derivatives, which regulate its transcriptional activity and participate in various biological processes such as cellular glucose and lipid metabolism, cell differentiation, proliferation and inflammatory responses [32]. However, saturated (SLCFAs) and unsaturated (ULCFAs) long-chain fatty acids exhibit distinct binding affinities for PPARβ/δ [11]. Mechanistically, FABP5 facilitates the nuclear translocation of PPARβ/δ agonists, particularly ULCFAs and retinoic acid, thereby promoting receptor activation. In contrast, SLCFAs, which do not serve as PPARβ/δ ligands, are not transported into the nucleus [11,12]. Non-targeted metabolomics revealed substantial metabolic alterations during the cyclic growth of yak HF, with Fatty Acyl compounds emerging as the most significantly changed metabolites during the transition to anagen and catagen. These findings demonstrate remarkable lipid metabolic remodeling in yak skin, aligning with proteomic evidence that shows pronounced changes in lipid and fatty acid-associated proteins. ULCFAs displayed synchronized expression patterns during both anagen and catagen, whereas SLCFAs showed inconsistent trends, suggesting that PPARβ/δ activation may be closely associated with yak HF cycling. HF growth depends on a precisely maintained balance in cell proliferation between the DP (mesenchymal component) and the HM (epithelial component) within the follicle. PPARβ/δ has been implicated in the regulation of proliferation and apoptosis across a variety of tissues and cell types [8,32,33]. GW501516, a selective PPARβ/δ agonist, stimulates human lung carcinoma cell proliferation [34]. PPARβ/δ knockout mice had 40% fewer skeletal muscle progenitors (satellite cells) than their wild-type littermates, and these satellite cells exhibited reduced growth kinetics and proliferation in vitro [35]. In psoriatic and atopic dermatitis skin, FABP5 is primarily localized to the nuclei of suprabasal keratinocytes, suggesting efficient local production of PPARβ/δ ligands that sustain its activation [36]. Interestingly, the levels of 5-HETE, arachidonic acid (ULCFA, PPARβ/δ endogenous ligands) are significantly increased compared to healthy skin [37,38]. This expression pattern closely resembles that of FABP5, PPARβ/δ, and ULCFAs observed in yak HF skin during anagen. We therefore hypothesize that PPARβ/δ is involved in the proliferation and apoptosis of HF cells, warranting further investigation.

PPARβ/δ was ubiquitously expressed in the nuclei of the DP, HM and outer root sheath in yak HF cycling. However, its expression exhibited marked dynamic changes, particularly within the DP and HM. Notably, DP expression of PPARβ/δ progressively declined as follicles transitioned from the growth (anagen) to regression (catagen) and ultimately the resting (telogen) phase, becoming nearly undetectable in telogen DP. To elucidate the functional role of PPARβ/δ, we isolated and cultured primary yak DPCs. The specificity of these cells was confirmed by positive immunostaining for the DP markers SOX2 and Vimentin [4,39]. Pharmacological activation of PPARβ/δ using the agonist GW0742 led to the downregulation of ACSL1, a rate-limiting enzyme in endoplasmic reticulum-associated lipid synthesis [28,29], suggesting a suppression of lipogenesis. Concurrently, we observed a significant reduction in the expression and immunofluorescence signal of the lipid droplet marker PLIN2 in both the cytoplasm and nucleus, indicating a decrease in intracellular lipid droplets [31]. Conversely, inhibition of PPARβ/δ with DG172 dihydrochloride yielded largely opposite effects, promoting lipid accumulation. These results demonstrate that PPARβ/δ is a key regulator of lipid metabolism in yak DPCs, modulating intracellular lipid content. This aligns with proteomic data suggesting PPARβ/δ-mediated lipid metabolic shifts occur in the DP in vivo. PPARβ/δ is a known promoter of fatty acid β-oxidation across diverse cell types [40]. For instance, its activation enhances mitochondrial biogenesis and lipid oxidation in the heart, improving function in heart failure models [41]. Similarly, in HIT-T15 cells, PPARβ/δ activation upregulates genes for fatty acid oxidation and mitochondrial uncoupling, boosting lipid catabolism and reducing apoptosis [34]. Our findings reveal a parallel role in yak DPCs. Importantly, PPARβ/δ inactivation resulted in substantial intracellular lipid accretion, implying that its loss of function may disrupt β-oxidation or potentiate lipogenesis. We subsequently investigated the functional consequences of these metabolic shifts on cell proliferation and apoptosis. Activation of PPARβ/δ, which reduced lipid content, correlated with enhanced DPCs proliferation. In contrast, PPARβ/δ inactivation, which induced lipid accumulation, was associated with suppressed proliferation. This suggests that PPARβ/δ influences DPCs proliferation by remodeling lipid metabolism. Given the dynamic expression of PPARβ/δ in the DP throughout the hair cycle, it is plausible that it modulates follicle growth via lipid metabolic regulation. It has been established that PPARβ/δ promotes fatty acid oxidation, thereby increasing ATP production. This enhanced cellular energy supply is likely a key mechanism by which it facilitates cell proliferation [42]. Conversely, lipid overload, potentially due to impaired β-oxidation, can lead to lipotoxicity, which may explain the observed decline in proliferation [43].

Lipids within the skin have been demonstrated to be closely associated with hair follicle growth in various animals, and lipid storage in yaks often changes due to environmental survival conditions [44]. From summer with abundant water and grass to the harsh winter with food scarcity, yaks typically undergo a process of lipid accumulation followed by depletion, which likely also occurs in their skin [22]. PPARβ/δ-mediated lipid metabolism may serve as one of the key pathways regulating lipid homeostasis in yak skin, thereby influencing the cycle of yak HF. However, the extent of changes in intradermal lipid metabolism across different skin tissues during the cyclical growth of yak hair follicles, as well as the influence of other lipid factors on this process, remains poorly understood. PPARβ/δ may represent only one of these factors, which constitutes a limitation of this study. In summary, these findings demonstrate that PPARβ/δ modulates lipid metabolism in yak DPCs, thereby influencing their proliferation. This mechanism may contribute to the regulation of cyclic HF growth in vivo.

## 5. Conclusions

This study provides the first demonstration that lipid metabolism undergoes significant alterations in the skin during the cycle of yak HF. Fatty acyl compounds, particularly ULCFAs, which exhibited a consistent trend of change, were identified as the most significantly altered metabolites during the transition of yak HF into anagen and catagen. PPARβ/δ was found to significantly regulate lipid changes within the DPCs of yaks and influence their proliferation and apoptosis levels. These findings suggest that PPARβ/δ may modulate the cycle of yak HF by altering lipid metabolism in the DP.

## Data Availability

The data supporting the findings of this study are detailed within the methods and Appendix A sections of the article. Additionally, TMT proteomics data and untargeted metabolomics data are available from the corresponding author. This dataset includes the raw sequencing data generated during the study.

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
