# Peer review of "The Role of PPARβ/δ-Related Lipid Metabolism in High-Altitude Adaptation of Yak Coat Based on Proteomics and Metabolomics"

_cells, 2025, doi:10.3390/cells14231843_

Round 1
Reviewer 1 Report
Comments and Suggestions for Authors
This study investigates the adaptive regulation of the hair follicle cycle in yaks living on the cold plateau of Tianzhu County at an altitude of 3100 m. The authors use proteomic, metabolomic, and cellular methods. In summary, PPARβ/δ likely regulates the hair follicle cycle in yaks by modulating lipid metabolism in the dermal papilla.
Abstract: It is suggested that the abstract be structured according to the sequence of experiments. You may use subheadings such as “Background/Objectives,” “Methods,” “Results,” and “Conclusions.”
The authors should also mention that they focused on the most enriched signaling pathway, but that other signaling pathways may also play a role. You may speculate on what further research might be needed to elucidate additional regulators of the hair follicle cycle for a cold plateau environment.
In this context, the question may also arise as to whether adaptation to cold and heat follows the same regulatory pathways or whether these are based on completely different mechanisms.
Limitations of the study should be mentioned.
Introduction:
Propose that authors add a schematic figure to illustrate the roles of PPARβ/δ.
Lines 48-51: please add references for these statements.
Lines 53 and following: There is a break in the text flow here because the connection to the (PPARs) remains here unclear. Please re-write this text passage. You must add the justification why the PPARs are important for the synchronized cyclic growth of a large number of hair follicles in yaks. This means you have to re-organize this section.
Lines 91-98: How many skin samples were taken and from which parts of the body.
Figure 2: captions with too small fonts, not readable.
Line 282: (A) 3. Please add legend.
Line 301-302 and 309-312: please move these sections to Discussion
Figure 3 A-E: captions with too small fonts, not readable.
Line 395-398: please this section move to Discussion
Discussion and Conclusions: the reader would expect that the authors come back to the main question raised in this paper: which specialized physiological mechanisms have you found that yaks can adap to such harsh environmental conditions.
Therefore, please make clear what are your new findings and which findings confirm previous hypotheses and results.
Then, you should explain the specific regulators in yaks which are important for the adaptation.
References: it looks that there went something wrong. Authors from 6-43. Information about the journal with volume and pages are often missing.
Author Response
|
Comments 1: Abstract: It is suggested that the abstract be structured according to the sequence of experiments. You may use subheadings such as “Background/Objectives,” “Methods,” “Results,” and “Conclusions.” |
|
Response 1: Thank you for pointing this out. I agree with this comment. Therefore, I have rewritten the abstract as requested. |
|
Comments 2: The authors should also mention that they focused on the most enriched signaling pathway, but that other signaling pathways may also play a role. You may speculate on what further research might be needed to elucidate additional regulators of the hair follicle cycle for a cold plateau environment. |
|
Response 2: Agree. This content has been incorporated into the Discussion section.
Comments 3: In this context, the question may also arise as to whether adaptation to cold and heat follows the same regulatory pathways or whether these are based on completely different mechanisms. Response 3: Agree. This content has been incorporated into the Discussion section.
Comments 4: Limitations of the study should be mentioned Response 4: Agree. This content has been incorporated into the Discussion section.
Comments 5: Propose that authors add a schematic figure to illustrate the roles of PPARβ/δ Response 5: Agree. A schematic diagram has been added.
Comments 6: please add references for these statements. Response 6: Agree. I have added.
Comments 7: Lines 53 and following: There is a break in the text flow here because the connection to the (PPARs) remains here unclear. Please re-write this text passage. You must add the justification why the PPARs are important for the synchronized cyclic growth of a large number of hair follicles in yaks. This means you have to re-organize this section. Response7: Agreed. Transitional sentences have now been added to improve the flow of the text.
Comments 8: How many skin samples were taken and from which parts of the body. Response8: Agreed. I have revised the paragraph on sample collection.
Comments 9: Figure 2: captions with too small fonts, not readable. / Line 282: (A) 3. Please add legend. / Line 301-302 and 309-312: please move these sections to Discussion. / Figure 3 A-E: captions with too small fonts, not readable. / Line 395-398: please this section move to Discussion. Response9: Agreed. The requested changes have been made.
Comments 10: Discussion and Conclusions: the reader would expect that the authors come back to the main question raised in this paper: which specialized physiological mechanisms have you found that yaks can adap to such harsh environmental conditions. / Therefore, please make clear what are your new findings and which findings confirm previous hypotheses and results. / Then, you should explain the specific regulators in yaks which are important for the adaptation. Response10: Understood. The content has been added to the final paragraph of the Discussion section as you recommended.
Comments 10: References: it looks that there went something wrong. Authors from 6-43. Information about the journal with volume and pages are often missing. Response10: The reference list has been reformatted as required. |

Reviewer 2 Report
Comments and Suggestions for Authors
The manuscript addresses an interesting question,namely the role of lipid metabolism alterations and regulation in the adaptations to cold environment of yak coat's periodic growth pattern . The experimental plan is well described, the methods are up-to-date. The results are clearly presented and interesting for the scientific community. Short description of the connection between the cold-warm weather and the hair follicle's phases could help to clarify the correlation with climatic environment of the sampled animals. The discussion beyond the interpretation of the results, might be extended to similar studies in other animals/plants which were adapted to survive in harsh environment.
Author Response
|
Comments 1: The manuscript addresses an interesting question,namely the role of lipid metabolism alterations and regulation in the adaptations to cold environment of yak coat's periodic growth pattern . The experimental plan is well described, the methods are up-to-date. The results are clearly presented and interesting for the scientific community. Short description of the connection between the cold-warm weather and the hair follicle's phases could help to clarify the correlation with climatic environment of the sampled animals. The discussion beyond the interpretation of the results, might be extended to similar studies in other animals/plants which were adapted to survive in harsh environment. |
|
Response 1: Thank you for pointing this out. I agree with this comment. The discussion primarily covers physiological and pathological studies in mice and humans, while research on other animal models is more limited. Thank you for your insightful comment. |
